# Impact of an Agriphotovoltaic System on Metabolites and the Sensorial Quality of Cabbage (*Brassica oleracea* var. *capitata*) and Its High-Temperature-Extracted Juice

**DOI:** 10.3390/foods11040498

**Published:** 2022-02-09

**Authors:** Hyeon-Woo Moon, Kang-Mo Ku

**Affiliations:** 1Department of Horticulture, Chonnam National University, Gwangju 61186, Korea; notorious931216@gmail.com; 2BK21 Interdisciplinary Program in IT-Bio Convergence System, Chonnam National University, Gwangju 61186, Korea

**Keywords:** cabbage juice, agriphotovoltaic, sensory evaluation, glucosinolate, glucosinolate hydrolysis, metabolites

## Abstract

To date, the impacts of agriphotovoltaic (APV) condition on the production yield of crop have been studied; however, the effect of APV production on the sensorial quality and consumer acceptability of the produce remains unexplored. Therefore, to address this knowledge gap, we cultivated “Winter Storm” cabbage under solar panels (20.16 kW) and in open field in 2020. The weight and diameter reduction rate of fresh cabbage grown under APV condition compared to open-field conditions were 9.7% and 1.2%, respectively. The levels of glucosinolates and their hydrolysis products were not significantly different in the fresh cabbage between the two conditions. The amount of volatile organic compounds, which may affect the perception of smell, were significantly higher in the cabbage juice prepared from the ones grown in open-field conditions than in the juice prepared from cabbages grown under APV conditions (*n* = 3, *p* < 0.01). However, untrained subjects could not distinguish the difference in the quality of the 2 sets of cabbage juices in the triangle test (*n* = 70, *p* = 0.724). Regardless of the distinguishing features of color, aroma, and taste, the subjects did not have any preference between the two different cabbage juices.

## 1. Introduction

As environmental problems, such as global warming caused by fossil fuel use, have emerged, renewable sources of energy have become an effective alternative. Several countries have announced climate-friendly energy policies [1]. South Korea announced “The Renewable Energy 3020” policy that aims to increase the proportion of renewable energy up to 20% of the total energy generation by 2030. To achieve this goal, agriphotovoltaic (APV) systems, which generate electricity from raised solar panels and allow crop cultivation under the solar panels simultaneously, are being actively developed.

According to the Japanese energy policy, crop yield under APV should not be less than 80% of those grown in open-field (OF) conditions to ensure food security [2]. Thus, securing food crop yield, and the arrangement of solar panels that allow cultivation, have become primary research topics. Efficiency of radiation use and productivity have been studied on lettuce grown under 2 types of PV systems (1.6 m and 3.2 m intervals) [3]. According to another report [4], there was no difference in crop yield of maize grown under APV and that grown under OF, whereas land equivalent ratio (LER), an indicator for estimating the productivity of land, was increased up to 2 times. Touil et al., [5] reported some horticultural crops’ yields were affected by shading under the APV system, and the production was found to be inversely related to the photovoltaic cover ratio. In addition to the impact of APV condition on productivity, a study by Weselek et al. [6] showed that the levels of heavy metals in crops grown under APV condition were similar to, or sometimes less than, those in crops grown in open-field conditions. Sensorial quality of crops grown under APV might be compromised because of changes in the microenvironment factors under APV system. Among the microclimatic variables measured, the solar radiation and mean soil temperature changed, whereas mean air temperature and relative humidity did not differ compared to that of OF [3,7]. Experiments have shown differences between OF and APV conditions in the levels of anthocyanin, antioxidant capacity, sugars, and organic acids in berries [8], and in color, firmness, and total soluble solids in tomatoes [9] and strawberries [10]. However, there is little information on the sensorial quality of crops grown under APV and on consumers’ acceptance. Without this information, consumers may not be willing to buy these crops, especially if they are not confident about their quality.

Raised solar panel in APV can change the microclimate because of its structure above the crops. The shaded part, caused due to the blocking of the sunlight by the PV panels, have decreased soil temperature and lower light intensity reaching the plants [11,12]. In general, an increase in shading leads to a decrease in plant yield. Several studies have shown that shading of plants not only affects their growth but also alters the composition or quantity of metabolites [13,14].

*Brassica* vegetables, including cabbage (*Brassica oleracea* var. *capitata*), contain anticarcinogenic phytochemicals, especially glucosinolates and its hydrolysis products [15,16]. In contrast to their health-promoting effects, they also have phytochemicals that adversely affects the taste or scent, [17,18]. Glucosinolates, sinigrin, and progoitrin have negative influence on consumers’ preference owing to their bitter taste [19]. Chiu et al. [20] suggested that indole-glucosinolate-derived hydrolysis products are associated with sensory attribution and preference in broccoli (*B. oleracea* var. *italica*). Changes in environmental factors including air temperature, soil temperature, and soil water content during growing season affect the content of glucosinolates, especially indolyl glucosinolates [21,22,23]. Therefore, the microclimate under the APV condition may alter the glucosinolate content. Furthermore, altered amount of glucosinolate might affect consumers’ preferences.

According to a report by the Ministry of Agriculture, Food, and Rural Affairs on food and dining statistics in 2020, the market share of vegetable juices in South Korea was 0.15 billion dollars. People began drinking cabbage juice because it was easy to consume and effective in recovery from stomach ulcers [24]. However, producing and storing the juice to ensure its freshness is difficult as it can easily degrade under the influence of microorganisms. The high temperature and pressure during food processing plays a role in sterilizing and preventing the formation of unnecessary components during storage by inactivating enzymes [25,26]. Therefore, juice produced under high pressure and high temperature could be stored in retort pouches for a long time.

Until now, research has been focused on the growth and yield of crops grown under APV condition or possible species selection [3,5,27,28,29,30]. However, sensory evaluation of crops and their processed product is poorly investigated. Therefore, this study aimed to investigate the effect of APV on sensorial quality of cabbage juice and whether it affects consumers’ acceptability.

## 2. Materials and Methods

### 2.1. APV System

An APV system was installed in the farm of Chonnam National University, Naju, Jeollanam Province, Korea (34° 58′ 28.4″ N, 126° 45′ 59.3″ E). The total area of the APV structure was 480 m^2^, with 3.3 m high columns (with a spacing of 4 m × 5 m). The specifications and layout of the PV panel were as follows: each panel had the length, width, and thickness of 1.68 m, 0.95 m, and 30 mm, respectively. The angle of panels was 35°. Four bi-facial panels were in four columns. The total power generation capacity of PV panels was 20.16 KW (Figure 1).

### 2.2. Collection of Microclimate Data

Microclimate data were collected using data loggers (ZL6, Meter Group inc., Pullman, WA, USA). We collected data on atmospheric and soil variables. Specifically, air temperature and humidity, solar radiation, and soil temperature and moisture were collected using ATMOS 14, PYR, and TEROS 11 sensors, respectively. ATMOS 14 and PYR sensors were installed 60 cm above the ground, and TEROS 11 sensors were installed 20 cm under the ground.

### 2.3. Cultivation of Cabbage

Winter cabbage (*B. oleracea* var. *capitata*, cultivar: “Winter Storm”) was purchased from Asia Seed Co. Ltd. (Seoul, South Korea), and sown on 27 July 2020, in 105-cell seedling trays. The seedlings were transplanted into 30 cm raised bed, covered with black plastic mulching film at 35 × 35 cm intervals, in the farm of Chonnam National University on 6 September 2020. NovaTec-Suprem fertilizer (Compo Expert, Münster, Westphalia, Germany) was applied at 100 g·m^−2^ to provide nutrients. Cabbages were harvested on 15 January 2021 to avoid extreme cold weather. In the OF and APV area, 5 cabbages were sampled and pooled per replication site (4 biological replications with 5 sub-sampling each). We weighed and measured the diameter of the cabbages using a scale and a ruler immediately after harvest. The diameter was measured at the longest axis of the width. A total of 5 fresh cabbages from each replication were chopped in quarters and freeze dried (MCFD8508, IlshineBioBase Co. Ltd., Dongducheon, Korea) at 80 °C and 5 mTorr after freezing with liquid nitrogen. Following freeze drying, the samples were ground and stored in a freezer at 20 °C until phytochemical analyses. Four biological replicates were used for the analyses of metabolites of ground cabbage powder.

### 2.4. Preparation of High-Temperature-Extracted Cabbage Juice

Cabbage juice was made by the Jangsoo Food Corp. (Naju, Korea) using the commercially employed method for producing cabbage juice. All harvested cabbages grown under open-field and bi-facial panels were pooled and juiced in a commercial juicer (above 110 °C, at 2 kPa pressure for 3 h) to mimic commercial-level, bulk production. The heated cabbages were slowly cooled down for over 30 min and then squeezed. The filtered juice was packed in retort pouches and sterilized at 100 °C for 20 min. Pouches were stored at 4 °C in a refrigerator until use. Three analytical replicates were used for the analyses of metabolites of cabbage juice.

### 2.5. Quantification of Glucosinolates

Glucosinolates in the cabbages were extracted and analyzed following a previously described method with little modification [31]. Freeze-dried powder (200 mg) and juice (200 µL) were treated with 2 mL of 70% methanol in a 15 mL tube at 95 °C for 10 min. The extract was cooled for 5 min and 500 µL of glucosinalbin (1 mM, isolated from *Sinapis alba* seeds) was added as an internal standard. The mixture was vortexed for 10 s and centrifuged at 1200× *g* for 10 min. The supernatant was decanted into a glass tube, and 1 mL of the supernatant was transferred to a 2 mL microcentrifuge tube. The lead barium acetate solution (150 µL, 0.5 M) was added to the tube, and the mixture was centrifuged at 15,000× *g* for 2 min. Following centrifugation, the mixture was drained into a poly-prep column filled with DEAE Sephadex A-25 (GE Healthcare, Piscataway, NJ, USA). When the mixture was passed, 3 mL 0.02 M pyridine acetate, 3 mL deionized water, and 500 µL sulfatase solution (20 U·mL^−1^, *Helix pomatia* Type-1, Sigma-Aldrich, St. Louis, MO, USA) were added to the column sequentially, but not at the same time. The desulfo-glucosinolates were eluted using deionized water and filtered through a 0.22 µm syringe filter. The sample was injected into a high-performance liquid chromatography (HPLC) (Agilent 1100, Agilent Technologies, Palo Alto, PA, USA) equipped with a Kromail RP-C18 column (250 mm × 4.6 mm; 5 µm; 100 Å) (AkzoNobel, Bohus, Sweden). The flow rate of mobile phase A (HPLC grade water) and mobile phase B (Acetonitrile) was 1.5 mL·min^−1^ with the following gradient system: 0 min, 1.5% B; 4 min, 4% B; 16.5 min, 25% B; 17 min 60% B; 19.1 min, 1.5% B; 24 min, 1.5% B. UV response factors were adapted to each glucosinolate for quantification [32].

### 2.6. Determination of Glucosinolate Hydrolysis Products and Volatile Compounds

Freeze-dried cabbage powder (75 mg) was mixed with 1.5 mL deionized water. The sample was incubated for 10 min to extract hydrophilic glucosinolates and myrosinase. After extraction, the powder mixture and cabbage juice (1 mL) were centrifuged at 15,000× *g* for 5 min. Following the centrifugation, 500 µL of the supernatant was transferred to a 1.5 mL PTFE tube. After which, 500 µL dichloromethane (DCM) with phenyl isothiocyanate (10 µg·mL^−1^) was added as a solvent for liquid–liquid extraction and internal standard, respectively. Then, the mixture was incubated at 37 °C for 4 h. After incubation, the mixture was shaken and centrifuged at 15,000× *g* for 3 min. The DCM layer was taken out into a vial with an insert, and 1 µL of the sample was injected into a gas chromatograph (Nexis GC-2030, Shimadzu, Kyoto, Japan) coupled to a gas chromatograph–mass spectrometer (GC/MS-QP 2020 NX, Shimadzu, Kyoto, Japan) and an autosampler with an injector (AOC-20i PLUS, Shimadzu, Kyoto, Japan). The capillary column (DB-5MS, Agilent Technologies, Santa Clara, CA, USA; 30 m × 0.25 mm coated with 0.25 µm film) was utilized for performing chromatographic separations. GC/MS conditions were as follows: injection and detection temperatures were set at 250 °C; flow rate of carrier gas was 1.5 mL·min^−1^; oven temperature was set starting at 35 °C reaching to 330 °C (increasing at the rate of 20 °C·mL^−1^); MS ion source and interface temperature were 300 °C and 250 °C, respectively; the mass scan range was 40–500 m/z.

### 2.7. Sensorial Evaluation

To assess the consumer acceptability of cabbage juice, we developed a testing methodology involving human subjects. This was reviewed by the Chonnam National University Institutional Review Board. Previously written book was referred for sensory evaluation [33]. Seven days before the experiment, research volunteer recruitment notices were posted throughout Chonnam National University Bulletins to recruit subjects. Total 70 subjects, 35 men and 35 women, participated in the evaluation. The number of subjects by age groups were 48 persons in 20 s, 8 in 30 s, 8 in 40 s, and 8 in 50 s and older. Cabbage juice pouches were opened, and the juice was poured into a bottle for homogenization. A total of 5 samples of 20 mL to each were prepared in jello shot cups; 3 for the triangle test and 2 for the attribute test. The sensory evaluation was conducted on 21 July 2021, in the lecture room of Chonnam National University, with the following procedure. First, untrained subjects evaluated sensorial characteristics (color, scent, and smell) of three samples and then chose one odd sample with a random number of three digits (triangle test). To hide sample identity and conduct sensorial evaluation without bias, three samples with three-digit numbers were divided into two groups, one with two OF samples and one APV sample (set A), and the other with one OF and two APV samples (set B). The sample configuration within each set was randomly assigned. Sets A and B were provided in 36 and 34 cases, respectively. Afterward, the samples in cups labeled with two different letters were evaluated by the subjects for their preference. The subjects had snacks and were given water for rinsing their mouths before evaluating the next set of samples. All subjects evaluated the samples under the same temperature and light environment (temperature—24 °C; light—700 Lux daylight LED). The questionnaire consisted of questions concerning whether subjects have had experience drinking cabbage juice (Experienced or Inexperienced), triangle test, and comparing the color, scent, and smell of the two samples. In addition, the subjects were asked to indicate their preference in the last. Attribution test result was collected as an ordinary seven-point hedonic scale compared with the control sample; we asked relative intensity of color, scent, and taste. We used Google Forms to create the questionnaire and its URL was converted to a quick response (QR) code. We provided the QR to the subjects to record their responses using their smartphones in a non-face-to-face way, accommodating for COVID-19 restrictions. The time spent on sensory evaluation was about 10–15 min per person.

### 2.8. Profiling of Primary Metabolites

Preparation of samples for profiling primary metabolites (lipids, organic acids, sugars, etc.) followed previously described methods with some modifications [34,35]. To sum up the process, freeze-dried cabbage powder and juice were extracted in methanol. Ribitol and tetracosane were used as internal standards of water-soluble and lipid-soluble compounds, respectively. Water-soluble and lipid-soluble compounds were divided into two phases by liquid–liquid extraction using deionized water and chloroform, respectively. Each organic phase was fully dried in a SpeedVac. Subsequently, methoxyamide (in anhydrous pyridine) was added in the tube containing dried water-soluble phase and incubated at 37 °C for 90 min with shaking at 800 rpm. For derivatization of the water-soluble metabolites, N-methyl-N-(trimethylsilyl)trifluoroacetamide and 1% trimethylchlorosilane (TMCS) were added, and the sample was incubated at 50 °C for 20 min with shaking at 800 rpm. It was added to N, O-bis(trimethylsilyl)trifluoroacetamide with TMCS for derivatization of the lipid-soluble metabolites. The mixture was incubated at 60 °C for 60 min with shaking at 800 rpm. The sample was transferred to vials with an insert and 1 µL was injected into a gas chromatograph (Nexis GC-2030, Shimadzu, Kyoto, Japan) coupled to a gas chromatograph–mass spectrometer (GC/MS-QP 2020 NX, Shimadzu) and an autosampler with injector (AOC-20i PLUS, Shimadzu). Chromatographic separation was performed by means of a capillary column (DB-5MS, Agilent Technologies, Santa Clara, CA, USA; 30 m × 0.25 mm coated with 0.25 µm film). The flow rate of the carrier gas (helium) was set to 1.2 mL·min^−1^. The three mass spectrophotometry parameters, ion source, interface temperature, and mass scan range, were set to 300, 250 °C, and 40–600 m/z, respectively. For analysis of water-soluble metabolites, the initial oven temperature was set at 80 °C for 2 min, then increased by 12 °C·min^−1^ to reach 330 °C, and kept at 330 °C for 5 min at the end. For analysis of lipid-soluble metabolites, the initial oven temperature was set at 150 °C for 1 min, then increased by 12 °C·min^−1^ to reach 320 °C, and kept at 320 °C for 7 min at the end. Identification of metabolites was processed based on the database of the National Institute of Standards and Technology (NIST).

For quantification of free amino acids, an analysis kit for GC/MS (EZ:faast, Phenomenex, Torrance, CA, USA) was used. Cabbage juice was added to 0.1 N HCl and amino acids were extracted at 25 °C for 10 min during shaking at 800 rpm. After then, amino acids in the supernatant were purified and derivatized according to the process delineated in the manufacturer’s manual. A capillary column (Zebron EZ-AAA amino acid GC, Phenomenex; 10 m × 0.25 mm) was used for separation. The oven temperature was kept at 110 °C for 1 min and increased to 320 °C at 26 °C min^−1^. The flow rate of the carrier gas was 1.5 mL min^−1^ and both injector and detector temperatures were set to 250 °C. The three mass spectrophotometry parameters, ion source, interface temperature, and mass scan range, were set to 240, 310 °C, and 45 to 450 m/z, respectively. Amino acids were compared to authentic standards for identification. Norvaline was used as the internal standard.

### 2.9. Determination of Color Difference of Cabbage Juice

Parameters for indicating color (*L**: lightness; a*: red to green; b*: yellow to blue) were measured by colorimeter (NR60CP, Shenzhen 3nh Technology Co., Ltd., Shenzhen, China). The equipment was calibrated with black (*L**: 0; a*: 0; b*: 0) and white (*L**: 97.13; a*: 0.05; b*: −0.76) boards. Cabbage juice was prepared in the same cup as the ones used for sensory evaluation, with a height of 5 cm from the surface. The sample from APV was used as a standard. The color difference (Δ*E*) was calculated using the following equation:$$\Delta E_{Lab}=\sqrt{{\left(\Delta \;L^{*}\right)}^{2}+{\left(\Delta \;a^{*}\right)}^{2}+{\left(\Delta \;b^{*}\right)}^{2}}$$

### 2.10. Data Processing and Statistical Analysis

The primary metabolites and amino acids data were normalized by appropriate internal standards for sample-wise normalization. The normalized data were used in principal component analysis (PCA) for obtaining a general outline of the variance of metabolites and information on differences in composition of metabolites among samples. Auto-scaling was applied to all variables for feature-wise (metabolites) normalization. Student’s *t*-test, *Chi*-square test, and descriptive statistics (95% confidence interval, CI) were conducted using Prism 5 (GraphPad Software Inc., San Diego, CA, USA).

## 3. Results and Discussions

### 3.1. Differences in Microclimate and Growth of Cabbage Grown under Open-Field Cultivation and Agriphotovoltaic System

#### 3.1.1. Changed Microclimatic Conditions under the Agriphotovoltaic System

In the OF, mean air temperature, maximum air temperature, mean soil temperature, and cumulative growing degree days (CGDD) were 0.2 °C, 0.2 °C, 0.5 °C, and 19.9 °C, respectively, which were higher than those under APV (Table 1). In addition, mean photosynthetic photon flux density (PPFD) per day under OF cultivation was 545.1 µmol·m^−^^2^·s^−1^ which was about 81% higher than the PPFD under APV. However, the differences between the systems in terms of mean air temperature and mean humidity were only 0.5 °C and 1.1%, respectively. Thus, one of the biggest differences in microclimate between APV and OF was sun radiation, which is consistent with the findings of a previous study [12].

#### 3.1.2. Cabbage Yield under Agriphotovoltaic System

The formation of the cabbage head grown under OF cultivation was slightly denser than that grown under APV system (Figure 2A), while there was no significant difference in size on the surface (Figure 2B). Weight of the cabbage grown under OF (1606 ± 504 g) was not significantly different (*p* < 0.05) from those grown under APV (1450 ± 315 g), while yield reduction in mean weight was 156 g. Weight loss rate of cabbage grown under APV compared with OF was 9.7%. Cabbage diameter grown in OF (21.3 cm ± 3.1) was not significantly different (*p* < 0.05) from those grown under APV (21.1 cm ± 1.9). According to a review by Touil et al., [5] regardless of the crop species, many experiments indicate that there are no significant shading effects by APV structure on crop yields.

### 3.2. Comparison of Metabolites between Grown under Open-Field and Agriphotovoltaic Systems

#### 3.2.1. Glucosinolates and their Hydrolysis Products

It has been widely understood that glucosinolates and their hydrolysis products are strongly associated with consumers’ preference and health effects (including quinone reductase inducing activity and various anticancer activities) of *Brassica* vegetables [18,20,36]. Therefore, to determine whether APV affects crop quality and human health-promoting compounds, glucosinolates and their hydrolysis products were measured. Glucosinolates in the cabbage powder of those grown under open-field (OF) and agriphotovoltaic (APV) systems were quantified. Total glucosinolates, total aliphatic glucosinolates, total indolyl glucosinolates, and individual glucosinolates were not significantly different in fresh cabbages between grown under OF and APV. Glucosinolates were not detected in cabbage juice (Table 2) because of thermo-degradation, and these findings are supported by a previous investigation [37], which showed 74.5% loss in total glucosinolates after 25 min of cooking. According to Oerlemans et al., [38] even though glucosinolates take time to break down, 3 h of heating during high-temperature extraction for juicing seems enough to break down. The major glucosinolate hydrolysis product including 1-cyano-2,3-epithiopropane (CETP), sulforaphane nitrile, and indole-3-acetonitrile were detected in all cabbage samples regardless of different growing conditions. Only nitriles or epithionitriles among the hydrolysis products were detected rather than isothiocyanates (ITC) or other hydrolysis product forms, indicating that cabbage has considerably high epithiospecifier protein (ESP) activity. This result corresponds to some of the previous studies, indicating that a majority of glucosinolates transform to nitriles or epithionitriles by hydrolysis in cabbage [39,40,41]. Furthermore, unlike the results of glucosinolates, hydrolyzed products were detected (Table 2), and were significantly different between OF and APV in juice (*p <* 0.05). However, high-temperature-extracted cabbage juice showed a completely different composition and level of glucosinolates hydrolysis products compared to freeze-dried cabbage powder.

#### 3.2.2. Difference in Primary Metabolites

In total, 24 metabolites were identified using NIST library or authentic standards (Appendix A). To visualize comprehensive metabolites differences between cabbage juice from OF and APV, principal component analysis (PCA) was performed. The 2D scores plot of fresh cabbage showed that there was no difference between OF and APV due to overlapping ranges within the 95% CI (Figure 3A). In contrast, the 2D scores plot of cabbage juice showed that OF and APV clusters were separated, with 59.9% of the total variation in primary metabolites explained by component 1 (Figure 3B). In addition, as per the results of *t*-test, glucose, fructose, and all identified lipid-soluble metabolites (palmitic acid, stearic acid, 2-palmitoylglycerol, 1-monopalmitin, glycerol monosterate) were significantly higher in OF than in APV, while malic acid, myo-inositol, sucrose, s-methyl-L-cysteine, and valine were significantly higher in APV than in OF. However, the difference in results between fresh cabbage and cabbage juice was caused by the difference between biological and technical replication, respectively. In case of commercial-level production, using pooled cabbage samples is a more realistic situation than processing cabbages from three biological replications separately. This is likely the reason for the difference in results between fresh cabbage and cabbage juice. In general, the method of biological replication has more fluctuation than technical replication due to deviation from various environmental factors. Therefore, analyzing products made by pooling cabbages from each treatment may be advantageous in comparing the quality differences of cabbage juice between the two systems.

### 3.3. Sensory Evaluation of Cabbage Juice

The triangle test was adopted to check whether people can distinguish the difference between the two samples (Figure 4). In this experiment, 30 of the 70 subjects were able to distinguish the odd sample from the other samples, but it was not a significant difference (*p* = 0.091). However, considering that half of panels received different sample configurations, there was no difference in set A (2 OF and 1 APV, *p* = 0.724) while there was a significant difference in set B (2 APV and 1 OF, *p* = 0.039). Among the subjects, 36 had experience in drinking cabbage juice, and those with experience distinguished the difference significantly (*p* = 0.034). These subjects, when asked how they were able to distinguish the odd sample out of the 2 OF and 1 APV samples, their answers were “strong taste”, “darker brown color”, and “overall strong feeling”. It was noteworthy that the subjects who received set B or have experienced cabbage juice before were able to discern; although, the difference was not significant with respect to the total number of people. Additionally, participants who distinguished set B gave higher scores in scent and flavor than set A (A: 4.15, 4.38; B: 4.76, 5.00). O’Mahony and Odbert., [42] showed that weak samples (water) followed by strong samples (saltwater) were better distinguished, but in the opposite order, the result was reversed. Accordingly, different sample configuration of batch might have affected the perception of the subjects in the triangle test.

The result of the attribute test showed that the average scores for color, scent, and flavor were 5.2, 4.5, and 4.9, respectively, based on a 7-point hedonic scale (Table 3). The CI of total mean of color, scent, and taste were 4.92–5.48, 4.19–4.78, and 4.52–5.20, respectively, suggesting color and taste were different. Subjects who had experience drinking cabbage juice (*n* = 36) scored 5.2, 4.7, and 5.1, with CI of 4.78–5.56, 4.29–5.05, and 4.60–5.56, respectively, suggesting color and taste were different. Subjects who had not experienced drinking cabbage juice (*n* = 34) scored 5.2, 4.3, and 4.5, and their CI was 4.79–5.64, 3.82–4.78, and 4.07–5.02, respectively, suggesting color and taste were different too. People who had experienced cabbage juice before tended to judge that the cabbage juice of OF is stronger than that of APV, than inexperienced people. However, regardless of the above results, preference between OF and APV sample was equal (34:34; 2 “no” responses; *n* = 70). This indicates that the difference observed by the subjects was not the critical level of difference for their preference.

### 3.4. Parameters Affecting the Sensory Properties of Cabbage Juice

#### 3.4.1. Volatile Organic Compounds

Volatile organic compounds from glucosinolate hydrolysis analysis are shown in Figure 5A. The amount of 2-methylpyrazine, 2,5-dimethyl pyrazine, dimethyl trisulfide, 1h-pyrrole-2-carboxaldehyde, and s-methyl methanthiosulphonate were significantly higher in the OF cabbage juice than in that of APV (*p* < 0.01, *n* = 3, technical replication). Pyrazines are found in roasted or toasted foods and are known to contribute to the roasted scent [43]. Dimethyl trisulfide was identified as a key aroma constituent of *Brassica* vegetables owing to its property to increase while cooking and attribute unpleasant scents [44]. In various cabbage cultivars, it was found that dimethyl trisulfide was detected and increased as extraction time goes by, whereas dimethyl disulfide was barely detected in some cultivars [45]. Therefore, most of the volatile compounds, higher in OF sample, might have influenced the participants, who could have sensed a stronger scent in sensorial evaluation.

#### 3.4.2. Differences in Colorimetric Effect

The value of Δ*E** was 1.11 ± 0.06 (Figure 5B). The observer was unable to differentiate when the range of Δ*E** was from zero to one, and an experienced observer was able to differentiate from one to two [46]. In this test, not only the experienced but also the other subjects judged the OF sample to be the one with the stronger color than the APV (5.2 score—4 indicates no difference; Table 3). Therefore, it was enough to notice color difference.

## 4. Conclusions

We investigated possible differences in the quality of cabbage grown under OF and APV and its high-temperature-extracted juice, empirically. There were no differences of yield and metabolites between cabbage grown under OF and APV. However, cabbage juice made by pooled samples from each treatment showed differences in metabolites and consumers’ response to one set of triangle test. Given together, bulked commercial-level production of cabbage juice may show difference in metabolites; although, this difference was not significant enough to alter consumers’ preference on the cabbage juice (Figure 6). To the best of our knowledge, this is the first report to test consumers’ preferences and sensorial differentiation of OF- and APV-grown produce. Different cultivation practice, cultivar selection, extraction process, and other factors may influence the quality of cabbage and its sensorial quality, suggesting more experiments are required. Nevertheless, this study will provide consumers and the scientific community with basic information on the quality of crops grown under the APV system.

## Figures and Tables

**Figure 1 foods-11-00498-f001:**
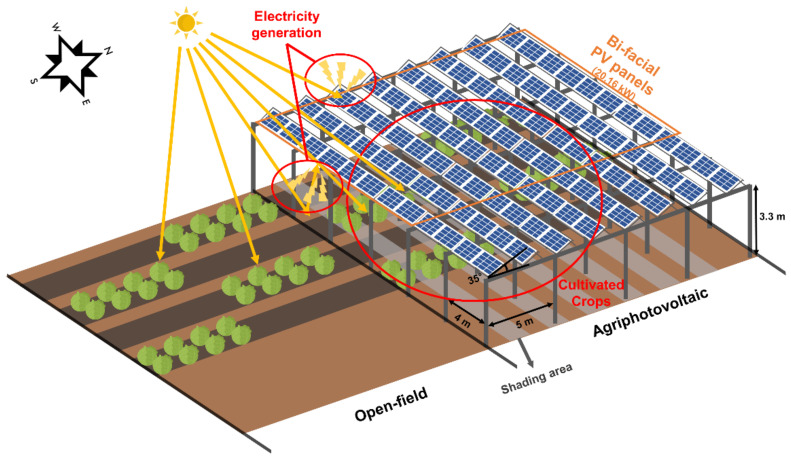
The conceptual diagram of agriphotovoltaic (APV) setup. Bi-facial photovoltaic (PV) panels generate power and crops are cultivated under the PV panels. The shading areas exist owing to sunlight blocked by PV panels.

**Figure 2 foods-11-00498-f002:**
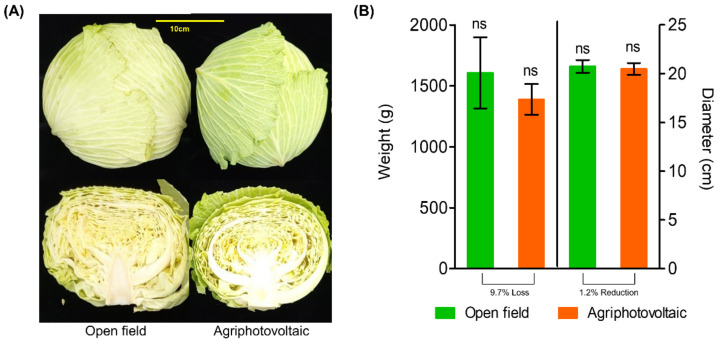
Pictures of cabbages (**A**); weight and diameter of cabbages grown under open-field (OF) and agriphotovoltaic (APV) conditions (**B**). Each parameter was measured immediately after harvesting (*n* = 4). The graph has a scale bar of 10 cm; ns—non-significant.

**Figure 3 foods-11-00498-f003:**
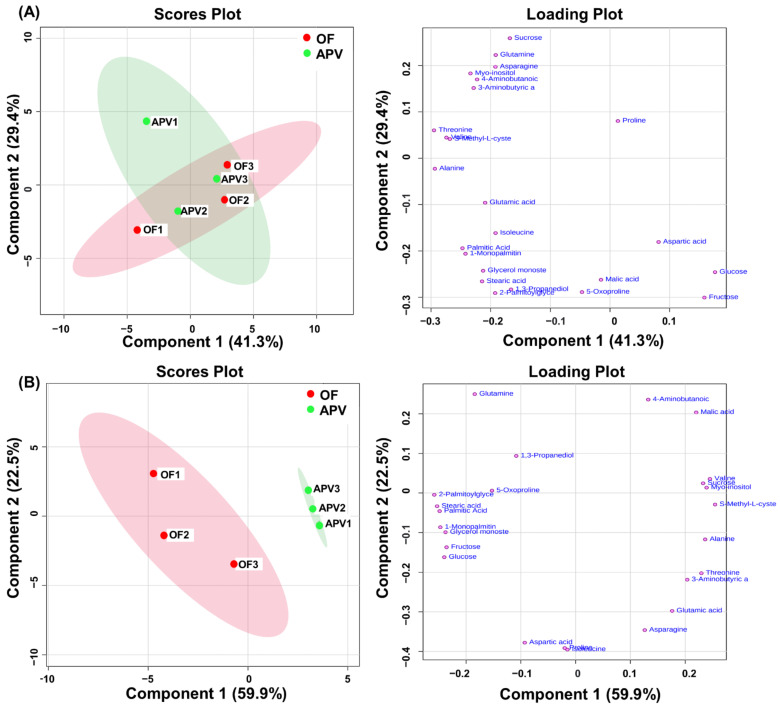
Scores and loading plots from principal component analysis (PCA) of fresh cabbage (**A**) and cabbage juice (**B**) from open-field and agriphotovoltaic systems. Metaboanalyst 5.0 was used for performing PCA.

**Figure 4 foods-11-00498-f004:**
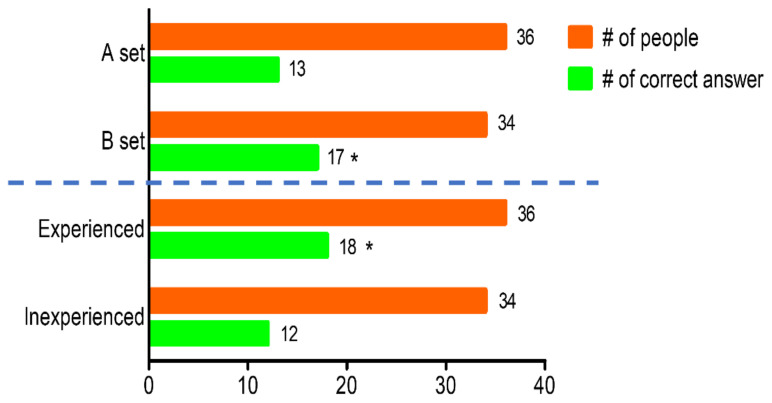
The number (#) of people who chose one sample as different from the other samples, and total number in each group. The data was collected by triangle test of cabbage juice made of cabbage grown under OF and APV (*n* = 70). Set A contained 2 OF samples and 1 APV sample, while set B had 2 APV samples and 1 OF sample. Asterisk (*) indicates significant differences by chi-square test (*p* < 0.05) between the total and the number of correct respondents.

**Figure 5 foods-11-00498-f005:**
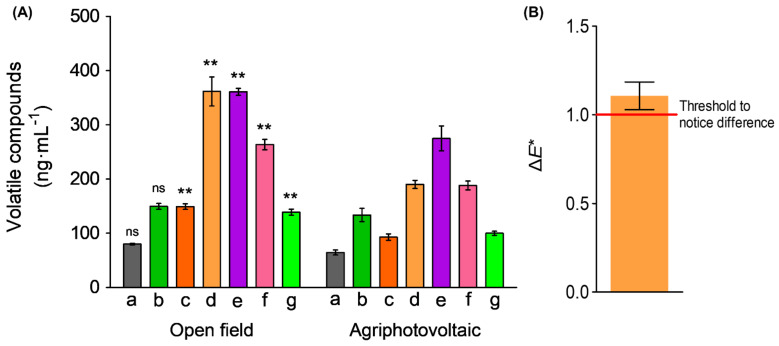
Volatile organic compounds (**A**) and color difference (Δ*E**) of cabbage juice OF when compared with APV (**B**). Different letters indicate the following: a—furanone; b—furfural; c—2-methylpyrazine; d—2,5-dimethyl pyrazine; e—dimethyl trisulfide; f—1h-pyrrole-2-carboxaldehyde; g—s-methyl methanthiosulphonate. Asterisks (* and **) indicate significant differences in Student’s t-test (*p* < 0.05 and 0.01, respectively, *n* = 3 technical replicates) between OF and APV. ns—non-significant.

**Figure 6 foods-11-00498-f006:**
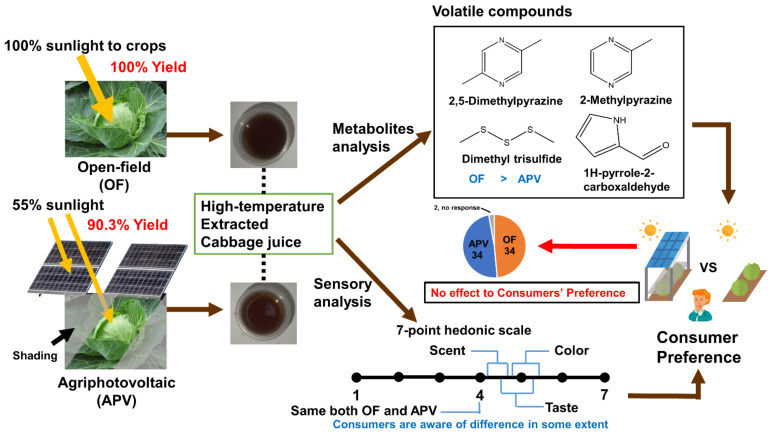
Overall scheme of the experiment. Cabbages were cultivated in conventional open-field conditions and under an agriphotovoltaic system. When the harvest of cabbage grown in OF was 100%, cabbage grown from APV showed a yield of 90.3%. Harvested cabbages were extracted at high temperatures (boiling). Metabolomic approach was utilized for both fresh cabbage and cabbage juice, but only cabbage juice was used for sensory evaluation. Some volatile compounds concentrations were higher in OF than APV. The result of the attribute test showed consumers are aware of the difference between the samples. However, it did not affect the overall consumer preference.

**Table 1 foods-11-00498-t001:** Microclimate information under open-field and agriphotovoltaic conditions.

Area	Mean Air Temperature (°C)	Max Air Temperature ^a^ (°C)	Min Air Temperature ^a^ (°C)	Mean Soil Temperature (°C)	CGDD ^a,b^ (°C)	Mean Humidity (%)	Mean PPFD per Day ^a^ (µmol·m^−^^2^·s^−^¹)
OF	9.2	32.9	−17.3	12.2	1670.9	54.4	545.1
APV	9.0	32.7	−17.1	11.7	1651.0	55.5	301.3

^a^ Max—maximum; min—minimum; CGDD—cumulative growing degree days; PPFD—photosynthetic photon flux density. ^b^ GDD = (Maximum day temp. + Minimum day temp.)/2—base temperature (5 °C).

**Table 2 foods-11-00498-t002:** Glucosinolates and their hydrolysis products in cabbages from open-field and agriphotovoltaic systems.

	Freeze-dried (μmol·g^−1^ DW)		Juice (μmol·mL^−1^)	
Glucosinolates	OF	APV	OF	APV
Glucoiberin	2.39 ± 0.37 ^a^	2.19 ± 0.38	n.d.	n.d.
Progoitrin	3.15 ± 0.57	2.89 ± 0.68	n.d.	n.d.
Glucoraphanin	2.70 ± 0.40	2.53 ± 0.47	n.d.	n.d.
Sinigrin	2.91 ± 0.40	3.03 ± 0.60	n.d.	n.d.
Gluconapin	0.65 ± 0.07	0.62 ± 0.09	n.d.	n.d.
Glucobrassicin	3.54 ± 0.31	3.52 ± 0.56	n.d.	n.d.
4-Methoxyglucobrassicin	0.80 ± 0.05	0.77 ± 0.03	n.d.	n.d.
4-Hydroxyglucobrassicin	0.44 ± 0.05	0.40 ± 0.05	n.d.	n.d.
Neoglucobrassicin	0.03 ± 0.01	0.02 ± 0.01	n.d.	n.d.
Gluconasturtiin	0.21 ± 0.03	0.23 ± 0.03	n.d.	n.d.
Total aliphatic GS	11.82 ± 1.75	11.26 ± 2.12	-	-
Total indolyl GS	4.84 ± 0.37	4.71 ± 0.58	-	-
Total GS	16.86 ± 1.79	16.21 ± 2.47	-	-
	Freeze-dried (μg·g^−1^ DW)		Juice (μg·mL^−1^)	
Glucosinolate hydrolysis	OF	APV	OF	APV
1-cyano-2,3-epithiopropane	55.81 ± 12.11	52.42 ± 10.47	n.d.	n.d.
1-cyano-3,4-epithiobutane	12.84 ± 0.91	10.44 ± 3.46	n.d.	n.d.
3-phenylpropanenitrile	7.05 ± 1.48	6.80 ± 0.74	0.49 ± 0.02	0.55 ± 0.03 *^b^
Erucin nitrile	2.34 ± 1.00	2.32 ± 1.19	0.73 ± 0.02 *	0.66 ± 0.03
Iberverin nitrile	1.27 ± 0.36	1.17 ± 0.59	0.44 ± 0.01	0.44 ± 0.02
Indole-3-acetonitrile	13.03 ± 2.22	14.30 ± 1.77	0.87 ± 0.08	1.00 ± 0.07 *
Sulforaphane nitrile	23.30 ± 4.04	21.03 ± 3.49	0.49 ± 0.06	0.51 ± 0.03
Iberin nitrile	n.d.	n.d.	0.34 ± 0.02	0.33 ± 0.01
Crembene	n.d.	n.d.	0.45 ± 0.04	0.46 ± 0.04
Goitrin	n.d.	n.d.	0.13 ± 0.02	0.12 ± 0.01
4-methoxyindole-3-acetonitrile	n.d.	n.d.	0.15 ± 0.02	0.16 ± 0.01
Total GS hydrolysis	115.63 ± 19.78	108.49 ± 15.30	4.09 ± 0.22	4.23 ± 0.18

^a^ The data were expressed as mean ± SD (*n* = 4, biological replicates, fresh cabbage; 3, technical replicates, cabbage juice). ^b^ Asterisks (*) indicate significant differences by Student *t*-test (*p* < 0.05) between OF and APV.

**Table 3 foods-11-00498-t003:** Attribute test of how OF sample differs from APV sample in color, scent, and taste.

	Total			Experienced ^d^			Inexperienced		
	Color	Scent	Taste	Color	Scent	Taste	Color	Scent	Taste
Mean ^b^	5.2 ± 1.2 ^a^	4.5 ± 1.2	4.9 ± 1.4	5.2 ± 1.2	4.7 ± 1.1	5.1 ± 1.4	5.2 ± 1.2	4.3 ± 1.4	4.5 ± 1.3
CI ^c^	4.92–5.48	4.19–4.78	4.52–5.20	4.78–5.56	4.29–5.05	4.60–5.56	4.79–5.64	3.82–4.78	4.07–5.02

^a^ Attribute scores were taken from 7-point hedonic scale how strong OF sample compare to APV sample. (1 = extremely weak, 2 = very weak, 3 = moderately weak, 4 = no difference, 5 = moderately strong, 6 = very strong, and 7 = extremely strong). ^b^ The mean data were expressed with ± SD (the numbers of total, experienced, and inexperienced participants were 70, 36, and 34, respectively). ^c^ The CI indicates 95% confidence interval of mean and expressed with lower CI–upper CI. ^d^ “Experienced” indicates those participants who have ever tasted cabbage juice before the experiment, and it was asked through the questionnaire during the sensorial test.

## Data Availability

Not applicable.

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
