# Peer review of "Impact of an Agriphotovoltaic System on Metabolites and the Sensorial Quality of Cabbage (Brassica oleracea var. capitata) and Its High-Temperature-Extracted Juice"

_foods, 2022, doi:10.3390/foods11040498_

Round 1
Reviewer 1 Report
1. Introduction
Line 30: correct the reference to the Korean Renewable energy; change “3020” by “2030”.
2. Materials and methods
2.4. Preparation of high temperature-extracted cabbage juice.
The handling of cabbages harvested under OF is not explained.
2.7. Sensory evaluation
This paragraph is quite confusing.
First, what does it mean “five samples of 20 mL”? There are two samples: juices from cabbages harvested under OF or PV conditions. The choice of the triangular test to determine whether or not there are sensory differences between the two treatments is correct but some clarification on the application of the test is necessary.
There are 6 different combinations of the two samples, but since there are only 70 judges available, not all combinations are tested the same number of times. First of all, it is surprising that set A has 36 trials and set B 34 when it should not be.
We must assume that the two samples have been evaluated in all possible combinations. If not, this should be explained in this paragraph.
Reference is also made to untrained judges, but what kind of experience the trained or "experienced" judges have is not explained. It is important to explain this, since it becomes relevant in the presentation of results.
Finally, the evaluation procedure through a QR and smartphones is not understood. Has all the sensory analysis been done this way or only the preference tests?
Since the judges are asked to evaluate the color of the juices, it is necessary to describe more precisely the conditions under which the sensory evaluations have been made.
2.8. Profiling of primary metabolites.
List at least the families of the so-called primary metabolites.
2.10. Data processing and statistical analysis.
Figure 2 seems more appropriate as Graphical Abstract, not relevant here.
Results and discussion
3.2.2. Difference in primary metabolites
Some information about concentrations is needed. This could be included as supplementary information. Also, references to supplementary material are missing.
3.3. Sensory evaluation of cabbage juices.
The general conclusion is that the judges do not find significant differences between the juices from the two types of cultivation. However, judges with experience in this type of product do detect differences between the juices, and also indicate that those made from OF cabbages have a greater intensity of smell. Could it be related to the higher concentration of volatile compounds in those cabbages? Further studies with trained panels could be interesting.
Table 3 could be completed showing the scores for OF and PV juices by experienced and non-experienced judges.
Author Response
We really appreciate for taking the time to review my paper. Please see the attached file. We answered your question and hope this was the answer.

Reviewer 2 Report
The paper “Impact of an Agriphotovoltaic System on Metabolites and the Sensorial Quality of Cabbage (Brassica oleracea var. capitata) and Its High -Temperature -Extracted Juice” contributes to the growth of literature for nutritionists as well as food producers offering plants products and on the quality of crops grown under the APV system.
Methods
The methods were adequately described. However, I would suggest to write:
Is the analysis based on a sample of selected consumers or from all over the country? The group of consumers is small in number (70 people), and therefore the characteristic of this group is significant - age, region.
What was the selection of the group?
The sensory tests were conducted in a sensory analysis laboratory equipped with individual booths (at controlled temperature and combined natural/artificial light), designed according to ISO standard?
What hours were the tests carried out?
Results
Whether the scale was structured or continuous?
If structured, the results should be the numbers of results should be averaged to a scale of 7-point.
Author Response

(The authors gave the same response as above.)
